# Design, Synthesis and Evaluation of New Multifunctional Benzothiazoles as Photoprotective, Antioxidant and Antiproliferative Agents

**DOI:** 10.3390/molecules28010287

**Published:** 2022-12-29

**Authors:** Riccardo Barbari, Chiara Tupini, Elisa Durini, Eleonora Gallerani, Francesco Nicoli, Ilaria Lampronti, Anna Baldisserotto, Stefano Manfredini

**Affiliations:** 1Department of Life Sciences and Biotechnology, Section of Medicines and Health Products, University of Ferrara, Via Fossato di Mortara 17-19, I-44121 Ferrara, Italy; 2Department of Life Sciences and Biotechnology, Section of Biochemistry and Molecular Biology, University of Ferrara, Via Fossato di Mortara 74, I-44121 Ferrara, Italy; 3Department of Chemical, Pharmaceutical and Agricultural Sciences, University of Ferrara, Via Luigi Borsari 46, I-44121 Ferrara, Italy

**Keywords:** benzothiazole derivatives, multifunctional, organic sunscreen UV-filter, antitumor, antioxidant

## Abstract

A current trend of research in the health field is toward the discovery of multifunctional compounds, capable of interacting with multiple biological targets, thus simplifying multidrug therapies and improving patient compliance. The aim of this work was to synthesize new multifunctional chemical entities bearing a benzothiazole nucleus, a structure that has attracted increasing interest for the great variety of biological actions that it can perform, and already used as a scaffold in several multifunctional drugs. Compounds are reported, divided into two distinct series, synthetized and tested in vitro for the antioxidant, and include UV-filtering and antitumor activities. DPPH and FRAP tests were chosen to outline an antioxidant activity profile against different radical species. The UV-filtering activity was investigated, pre- and post-irradiation, through evaluation of a O/W sunscreen standard formulation containing 3% of the synthetic compounds. The antitumor activity was investigated both on human melanoma cells (Colo-38) and on immortalized human keratinocytes as a control (HaCat). A good antiproliferative profile in terms of IC_50_ was chosen as a mandatory condition to further investigate apoptosis induction as a possible cytotoxicity mechanism through the Annexin V test. Compound **BZTcin4** was endowed with excellent activity and a selectivity profile towards Colo-38, supported by a good antioxidant capacity and an excellent broad-spectrum photoprotective profile.

## 1. Introduction

Nowadays, one of the most difficult challenges in the medicinal chemistry field is to tackle multifactorial diseases. These are defined as illnesses characterized by a heterogeneous etiology (ranging from environmental factors to genetic mutations) and variegate symptoms (which could involve numerous and different biological targets). Multifactorial diseases include Alzheimer’s, atherosclerosis, CNS disorders, rheumatoid arthritis, asthma, and cancer [1]. In order to find a suitable therapy, the use of a single therapeutic agent directed toward a specific molecular target has proven to be inefficient and prone to the development of drug-resistance phenomena [2].

The most common approach nowadays is a “drug cocktail” with two or more actives belonging to several therapeutic classes. However, this approach is difficult due to undesired drug–drug interactions, different pharmacokinetics, toxicity, bioavailability and costs of combination therapy; novel strategies are therefore continuously sought to overcome these drawbacks [3]. One of the new frontiers is the discovery of a single molecular entity endowed with at least two (or more) biological actions [4,5,6], with fewer side effects and thus overall better therapeutic index. Those “multifunctional drugs”, as they are defined, are able to act on different sites of action at the same time, to lower the undesired side effects and to increase the general compliance of the patient [7,8].

Skin cancer is one of the hardest challenges in the field of multifactorial disease: it is the world most incident malignancy [9], with over 2 million new cases discovered every year. As a matter of fact, for every three diagnoses of cancer, one involves skin cancer and the incidence is expected to increase more and more due to depletion of the ozone layer in the stratosphere, although the major risk factors are still related to UV ray exposure, sunburn histories, phenotype and genetics [10].

It is known that the wrong or excessive exposure to UV rays may result in oxidative stress, which comes from cellular accumulation of reactive oxygen species (ROS) not balanced by the cellular antioxidant defense, is the main causes of both melanoma and non-melanoma skin carcinoma [11]. As proof, a significant overexpression of antioxidant enzymes (copper-zinc superoxide dismutase, manganese superoxide dismutase and catalase) was reported in human melanoma biopsies when compared with surrounding non-tumor tissue [12]. Although the effectiveness of certain sunscreens in preventing at least some forms of skin cancers is quite established, none of those currently available seem to contrast UV exposure side effects, in particular against UVA-induced free radical generation. It remains a challenge to identify new sunscreen UV filters that could provide both UVA and UVB protection and, at the same time, operate as quenchers towards possible formation of ROS and other free radicals.

It is also well-known that a wide variety of polyphenols or phytochemicals, belonging to the superfamily of natural flavonoids, were reported to possess substantial skin photoprotective effects (antioxidant, anti-inflammatory and anticancerogenic capacity) and also work as weak UV filters, due to their general structure and the presence of hydroxyl and aromatic moieties which make them able to absorb UV light in a wide range of wavelengths [13]. Recent research suggests that polyphenols could be an effective skin protection source from the effects of UV radiation (UVA and UVB) [14]. Application and consumption of polyphenols was shown to lead to lower UVB-caused skin sunburn.

PBSA (2-phenyl-1-H-benzimidazole-5-sulfonic acid) (Figure 1) is a heterocyclic UVB sunscreen filter widely used in the past two decades due to its effectiveness, safety and optimal water solubility. However, this molecule does not have the capacity to provide an extensive, broad-range UV protection, and needs to be used in a suitable combination with other organic or inorganic filters whose photoprotective activity is shifted towards the UVA range, as, for example, butyl methoxydibenzoylmethane (avobenzone, organic) or titanium dioxide (inorganic). Moreover, PBSA does not possess any antioxidant activity.

Our previous results [15] indicate that the insertion of hydroxy group on the phenyl moiety of PBSA led, in some cases, to new more potent UV sunscreen filters also endowed with an excellent radical scavenging and antioxidant capacity. It is worth mentioning that these new molecules also boosted the activity of commercial sunscreens when used in suitable combinations.

In our past studies we also investigated the possibility of bioisosteric replacement of the heterocyclic nucleus [16,17,18] and the insertion of a new acyl-hydrazone linker between the heterocycle and the aromatic ring to obtain newly synthetized benzofuran-hydrazones, indole-hydrazones and benzimidazole-hydrazones [19,20,21], in order to reach a structure which could retain or even increase the photoprotective features of PBSA and express antioxidant and/or antiproliferative capacity as well. Some of the compounds previously obtained presented an excellent wide range photoprotective capacity coupled with very good antioxidant activity, a dualism which enables those molecules to protect against both UVA/UVB and oxidative stress, both known to be directly related to DNA damage and onset of skin tumors [22].

Encouraged by the results, we obtained 2-arylbenzothiazoles substituted with one or more hydroxyl moiety in terms of multifunctional activity, with some endowed with UVA filtering properties, antioxidant capacity and an antiproliferative effect on tumor cells [18], we decided to undertake the design and synthesis of a new series of compounds by incorporating both the benzothiazole nucleus and the acyl hydrazone linker discussed above (Figure 1).

In this work, we designed, synthesized, and evaluated a set of 11 molecules (some already known, some newly synthesized) bearing the benzo[d]thiazole scaffold; for six of them, we considered the acyl-hydrazone derivatives previously published and decided to apply the same linker to the benzothiazole nucleus, leading to the compounds **BZTidr1–6**. For the remaining five, we designed and applied a new linker, which mimics the acyl-hydrazone moiety and retains the π conjugation, to the benzothiazole nucleus, leading to the cinnamidic derivatives **BZTcin1–5**. This allowed us to connect 2-aminobenzo[d]thiazole to cinnamic acid derivatives, for example, for coumaric or caffeic acid, whose antioxidant properties are well documented [23]. It should be noted that, derivatives of 4-methoxycinnamic acid, such as octinoxate and amiloxate, are included in the list of suitable UV-filters for a cosmetic formulation, as indicated in the VI Annex to Regulation (EU) n°1223/2009 and are currently among the most popular UVB-sunscreens in commercial products.

The compounds of the two series can be considered specular, as they carry the same substituents on the aromatic ring and differ only in the nature of the linker itself. This allowed us not only to compare the biological activity of the two types of linkage, but also to present a preliminary SAR about the effects of nature and position of the moieties on the aromatic ring. In order to investigate their potential multifunctional profile, each compound was tested for the determination of the antioxidant, UV-filtering and antiproliferative activity.

## 2. Results and Discussion

### 2.1. Chemistry

In this work, we synthesized 11 compounds: six of them incorporate an acyl-hydrazone linker, and five were obtained from the coupling of commercially available 2-aminobenzo[d]thiazole with cinnamic acid derivatives.

For the synthesis of **BZTidr1–6**, we followed and adapted an already reported procedure for the synthesis of a benzoimidazole-2-carbohydrazide derivative [19]: the initial step was to achieve the intermediate benzo[d]thiazole-2-carbohydrazide (**2**), which was obtained by treating the starting compound ethyl-2-benzo[d]thiazole-carboxylate (**1**) with a solution of hydrazine hydrate, and heating to reflux for approximately 1 h, in ethanol. Then, compound **2** was refluxed with the appropriate aromatic aldehyde for 3–8 h, in order to achieve the acyl-hydrazone derivatives **BZTidr1–6** (Figure 1). According to literature, a singlet downfield resonating signal in the range of 8.30–9.49 ppm and related to the -N=CH- proton, confirms that only the E-isomer was obtained [24].

For the synthesis of **BZTcin1–5** the aim was to generate a new amide linker between the amine function of 2-amino-benzo[d]thiazole (3) and the carboxylic moiety of the appropriate cinnamic acid derivative. Among different activating system combinations (DIC/HOBt, EDC/HOBt, DCC/DMAP, DCC/NHS), EDC/HOBt was the most suitable. However, at the beginning, the yield was not satisfying, and we tried to refine the procedure through a variation of several conditions, such as temperature, solvent and time of the reaction. The best conditions, then taken as a model for the entire series, included activation of carboxylic acid with a mixture of EDC·HCl (1-Ethyl-3-(3-dimethylaminopropyl)carbodiimide hydrochloride) and HOBt (1-hydroxy-benzotriazole), CH_3_CN as solvent, and overnight stirring at 55–60 °C (Figure 2). This was found to be the best compromise to enhance the reactivity of the aromatic amine and to limit side reaction involving phenolic moieties.

### 2.2. Antioxidant Profile

In recent years, antioxidants and their therapeutic potential in the prevention and modulation of oxidative damage have assumed a central role, and this has prompted an increasing interest in the search of new molecules, natural and synthetic, to counteract the course of disorders associated with excess free radicals in the body.

There are several in vitro investigation methods for measuring antioxidant capacity: the different methods allow the analysis of the reactivity of molecules toward different types of radicals, depending on the nature and mechanism of action of the antioxidant species.

The evaluations were conducted by the DPPH test and the FRAP test (Table 1).

The DPPH assay is widely used, and the simplest and fastest in vitro method for evaluating the antioxidant power of extracts or isolated compounds; it is based on the depletion of the 2,2-diphenyl-1-picrylhydrazyl radical (DPPH˙) by an antioxidant molecule with scavenger activity. This is an extremely sensitive test that can detect the activity of active ingredients present in very low concentrations.

The FRAP (Ferric Reducing Antioxidant Power) test allows the measurement of the ability of antioxidants to reduce ferric ion to ferrous. This method represents one of the simplest, quickest and least expensive methods for the determination of reducing properties on oxidized species in vitro.

For the DPPH test, the synthesized compounds were first tested at the concentration of 1 mg/mL, to evaluate the percentage of inhibition of the DPPH radical. Subsequently, only the derivatives that showed a percentage of inhibition greater than 75% were selected and further investigated in terms of IC_50_ (µg/mL).

Regarding the results obtained with the DPPH assay for the **BZTidr1–6** series, it is interesting to note that the compound **BZTidr4**, which has a catecholic functionality, displayed great antioxidant activity, with an IC_50_ equal to 3.27 µg/mL. The presence of two adjacent aromatic hydroxyls is generally associated with an excellent antioxidant capacity due to the ability to form a stable radical thanks to the effect of resonance. **BZTidr2** and **BZTidr3**, with a hydroxyl in para and a methoxyl in para, respectively, did not show significant antioxidant activity, as did the derivative **BZTidr5** (hydroxyl in meta and methoxy in para). Interestingly, **BZTidr6**, which is the structural isomer of **BZTidr5**, has instead shown a good antioxidant activity, with an IC_50_ equal to 26.22 µg/mL. These results suggest that a single hydroxyl in the para position is not sufficient to exert a satisfactory inhibitory activity against the radical, as shown by the low activity of **BZTidr2**. However, when an additional hydroxyl or a methoxy is found in the ortho position of this hydroxyl, the inhibitory activity towards DPPH increases considerably.

The results relating to the **BZTcin1–5** series confirm the above reported results. The **BZTcin4** derivative showed the strongest inhibitory activity of the DPPH radical, with a percentage equal to 92.54%, and an IC_50_ equal to 4.86 µg/mL. **BZTcin5** also showed slight antioxidant activity at the concentration of 1 mg/mL with an inhibition rate of 40.15%. The results confirm that the greater activity is related to the presence of a catechol moiety. Considering the derivative **BZTcin2**, characterized by a hydroxyl in the para position, we observed that this single hydroxyl is unable to confer a satisfactory antioxidant activity, as well as in the case of a methoxyl always in the para position (**BZTcin3**). However, the antioxidant capacity was slightly increased by inserting a hydroxyl in an ortho position compared to methoxy (derivative **BZTcin5**).

In regard to the FRAP test, the results of the **BZTidr1–6** series were similar to those of the DPPH, showing a better activity profile for the derivatives **BZTidr2**, **BZTidr4**, **BZTidr5**, and **BZTidr6**. The compound **BZTidr4** (with two hydroxyls in ortho between them) was found to be the most effective of the series, with an excellent antioxidant activity, confirming the relationship between antioxidant capacity and the number of hydroxyl functions on the benzene ring. For the **BZTcin1–5** series, the derivatives **BZTcin4** and **BZTcin5** were the most effective, with a good antioxidant activity, again confirming the importance of substituents on the benzene ring; in fact, in the FRAP test, the compound containing two hydroxyls in the 3,4 position respectively, was found to be the most active (**BZTcin4**), with a value of 4583.10 µmolTE/g.

### 2.3. Evaluation of Filtering Properties

We next investigated the protective potential against UV radiation of each compound. Specifically, in vitro tests were conducted to determine the parameters related to absorption in the UV spectrum and those fundamental to the relative evaluation of the filtering potential and thus candidature as possible UV sunscreen ingredients of the candidates under examination, including λ critical and SPF (sun protection factor). Since the SPF is correlated to the UV quantitative absorption of the substances, the wavelength of maximum absorption (λmax) and the molar extinction coefficient (ε) of the derivatives in solution (MeOH-DMSO) were evaluated (Appendix A). The UV spectra of the compounds **BZTidr1–6** and **BZTcin1–5** were recorded in the UVA-UVB range (290–400 nm) to evaluate their profile and compare it to that of the PBSA, a commercial UVB filter that has a λmax of 302 nm. After this peak, the absorbance of PBSA drops, reaching almost zero in the 325 nm range, and thus not providing absorption in the UVA region.

From the UV spectra recorded for the hydrazone series (Appendix A), it was evident that the λmax of **BZTidr1**, the only compound without substituents on the aromatic ring, was the lowest in the series. The λmax, on the other hand, increased as the number of hydroxyl and methoxy substituents increased; in fact, these groups can be defined as auxochromic, that is, capable of extending the conjugation of the molecule causing a bathochromic shift, in this case towards UVA. In fact, **BZTidr4**, bearing two hydroxyls on the aromatic ring in an ortho position, has the highest λmax of the entire series (346.2 nm). It was also observed that as the number of substituents on the ring increased the absorbance at λmax decreased. In fact, **BZTidr2** and **BZTidr3** (monosubstituted) and especially **BZTidr1** (unsubstituted) have higher absorbance values than disubstituted compounds, an effect defined as hyperchromic.

From the UV spectra recorded for the cinnamide series (Appendix A) it was found that the λmax of **BZTcin1** (370.0 nm), the only compound in the series that does not have substituents on the benzene ring, has the lowest value; the λmax of the other compounds show increasing values based on the number of substituents –OH and –OCH_3_, auxochromes that lead to a bathochromic shift of the absorption spectra. **BZTcin4**, which has the catecholic portion with the two adjacent hydroxyls on the benzene ring, show the highest λmax compared to the entire series (347.7 nm). The spectra show that all the λmax of cinnamic derivatives are shifted to wavelengths greater than the λmax of the PBSA (302 nm). Additionally, for the spectra of this series, an absorption difference was observed between the monosubstituted and disubstituted systems as described above.

Once the λmax of each derivative was established, the molar extinction coefficient ε was calculated by applying the Lambert-Beer equation (Appendix A): the higher this value, the greater the ability of the compound under examination to absorb UV radiation. All compounds of the **BZTidr1–6** series exhibited values comparable to or higher than those shown by the reference UVB filter (PBSA). The excellent values of **BZTidr1**, **BZTidr2** and **BZTidr3** are ascribed to the aforementioned hyperchromic shift. In regard to the **BZTcin1–5** series, the molar extinction coefficients calculated were very high, and the best values were those of **BZTcin1**, **BZTcin2** and **BZTcin3**, also attributable to the hyperchromic shift of the absorption spectra. For both series, the excellent absorption capacity of ultraviolet radiation can be traced back to the high π-type conjugation that characterizes them.

#### 2.3.1. In Vitro Photoprotective Activity of Sunscreen Formulation Containing Benzothiazole Derivatives

All synthesized compounds were formulated at 3% and their photoprotective efficacy, determined by spectrophotometric measurements using the modified Diffey-Robson method [25], was compared with that of the lead compound PBSA (used at 3% in the same formulation) and using the base without active as a control (Table 2).

According to the Food and Drug Administration [26], a broad-spectrum sunscreen, capable of simultaneously protecting the skin from UVA and UVB rays, should have an λ_c_ value greater than 370 nm. On the basis of this criterion, all compounds in both series met this request.

Although all derivatives are classified as broad spectrum, the SPF value was remarkable only for **BZTcin2** (SPF 12.26), with a hydroxyl in the para position, and **BZTcin4** (SPF 9.70), bearing a catechol moiety on the benzene ring, and, at the same time, these derivatives showed the highest UVAPF0 values (11.79 and 11.40, respectively). The UVA/UVB ratio, as defined by the 2006 EU recommendation (2006/247/EC), should be at least 1/3. All compounds show a UVA/UVB ratio greater than or equal to one and met this requirement, showing absorption mainly in the UVA region and confirming the best profile for the **BZTcin2** and **BZTcin4** derivatives. The derivative **BZTcin5**, which carries a methoxyl with a hydroxyl in ortho on the aromatic ring, also showed an interesting profile, with values of SPF and UVAPF0 doubled and tripled with PBSA at 3%.

It should be emphasized that none of the derivatives of the hydrazine series provided noteworthy SPF and UVAPF0 values, suggesting the contribution of the linker to the UV filtering activity.

#### 2.3.2. Photostability Study

To outline the efficacy and safety profile of a sunscreen formulation, it is essential to provide data relating to photostability. It is possible to provide photodegradation data by recording, pre- and post-irradiation, the transmittance of PMMA plates with finger-coat formulations subjected to a dose of UVA capable of causing erythema, according to the criteria of Garoli et al. [27]. By means of Equations (2) and (3) it was possible to evaluate the residual percentage of SPF in vitro and of UVA-PF for all the prepared formulations. Of greatest interest are the values obtained for the derivatives **BZTcin2** and **BZTcin4**. For the two compounds, the % SPFeff value was between 86.91% (**BZTcin4**) and 96.00% (**BZTcin2**), while the% UVA-PFeff value was between 85.50% (**BZTcin2**) and 93.25% (**BZTcin4**), indicating the two synthetic compounds as photostable; in accordance with the regulatory aspect, in fact, a filter can be considered photostable if, after irradiation, it shows a residual percentage of SPF in vitro and UVA-PF greater than or equal to 80 [28].

### 2.4. Antiproliferative Activity

All synthesized compounds were tested in order to evaluate their in vitro antiproliferative activity. The human melanoma Colo-38 cell line was selected as a tumor skin model, while the immortalized human HaCaT keratinocytes were chosen as a control for the evaluation of the cytotoxicity towards non-cancerous cells and of the selectivity towards tumor cell lines. Data were summarized in Table 3 and the antiproliferative activity was expressed as the IC_50_ value in µM (only active compounds are shown; compounds not reported did not explicate any activity towards the considered cell lines below 100 µM concentration). The selectivity index (SI) was also calculated and is the ratio between the IC_50_ of a given compound towards the non-cancerous cell line and the IC_50_ of the same compound towards the tumor cell line.

Cisplatin was used as positive control on HaCat cells (IC_50_ = 2.0 ± **0.40** µM), while hydrazones, such as N^1^-(4-arylidene)-1H-(2-OH-4-N(Et)_2_-phenyl)-[*d*]imidazole-2-carbohydrazides (IC_50_ = 0.7 ± 0.06 µM), were used on melanoma Colo-38 cells. As shown in Table 3, the **BZTidr** series, with the exception of **BZTidr2** and **BZTidr4**, did not express any antiproliferative activity at all. Those two compounds were able to inhibit 50% of Colo-38 cell line proliferation but only at very high concentrations (92.61 and 92.85 µM, respectively). The SI was estimated to be higher or equal to 1.08 as those compounds were not investigated above 100 µM concentration.

Compounds from the **BZTcin** series show a stronger activity against the melanoma cell lines; in particular, **BZTcin2**, **BZTcin4** and **BZTcin5** had IC_50_ values in the low micromolar range (35.18, 8.31 and 10.00 µM respectively). This led us to the hypothesis that the cinnamidic linker could play a significant role in the antitumor effects against Colo-38, due to the nature of the substituent in position 3 and 4. In fact, **BZTcin2** and **BZTcin4** both present a hydroxyl group in the para position. The insertion of a second hydroxyl group in the meta position seems to increase the inhibitory activity (as confirmed by the inhibitory growth effect of **BZTcin4**, over 4-fold better than **BZTcin2**), while the replacement of the hydroxyl group in para with a methoxyl moiety does not seem to affect this activity. As far as selectivity is concerned, it was reported that a SI equal or superior to 3 is auspicable in order to consider an active potential antitumoral candidate [29]. Among the compounds tested, **BZTcin2** comes closer to this criterion with an SI value of 2.84 toward Colo-38 cell lines with respect to HaCat cell lines. Unfortunately, **BZTcin4** and **BZTcin5** were the only two of the series to also exert an inhibitory growth effect on the HaCat cell line. Despite this, the compounds still show selectivity towards melanoma cells with SI values of 4.59 and 8.33, respectively; those values, taken together with their IC_50_, reflect their potential as antitumor agents, both in terms of potency and selectivity.

It is also noteworthy that, with the exception of **BZTcin4** and **BZTcin5**, all other compounds tested show no antiproliferative effects on a healthy cell line. This non-toxicity profile towards HaCat keratinocytes is per se a remarkable result and confirms the multifunctional profile of those compounds, which are supposed to target only tumoral cell lines.

### 2.5. Pro-Apoptotic Activity

In order to further investigate the antiproliferative activity, a study of apoptosis on Colo-38 and HaCat cell lines was carried out. Only benzothiazole derivatives which have shown noteworthy antiproliferative activity (IC_50_ < 100 µM) were subjected to the Annexin V release assay.

The Annexin V assay was conducted at two concentrations approaching IC_50_ values toward the selected cell lines reported in Table 3. The complete set of data is reported in Table 4, while representative data from the analysis of the corresponding Muse Analyzer plots are shown in Figure 2.

Among the tested benzothiazole derivatives, a lack of apoptosis induction towards the healthy cell lines was observed, as the percentage of cells in the early- and late-apoptosis stage does not differ significantly from the percentage of the same population treated with the vehicle (MeOH). It can be concluded that there is no significant apoptosis induction towards the HaCat cell line.

We report that **BZTcin2** and **BZTcin5** exhibit the best pro-apoptotic activity of the series, showing total apoptosis values over 30% and 20%, respectively. This effect seems to be dose-dependent as far as **BZTcin2** is concerned, while **BZTcin5** did not shown the same correlation pattern. Taken together, these data show that apoptosis could be a possible cytotoxicity mechanism for those compounds; however, due to the low percentage of apoptosis induction efficiency, it is also non-prominent. It should also be noted that **BZTcin4**, which showed the best IC50 value towards Colo-38, does not induce apoptosis at any of the concentrations tested. Further studies in this direction are required in order to evaluate other possible cytotoxicity mechanisms.

## 3. Materials and Methods

### 3.1. General

All reagents were purchased from Sigma Aldrich SRL, Milano, Italy, except for 2-aminobenzo[d]thiazole which was purchased from Acros Organics, Geel, Belgium. All solvents used were purchased from Carlo Erba Reagents SRL, Milano, Italy, and used without further purification. Silica gel plates were used to perform TLC analysis (Macherey-Nagel Poligram SIL G/UV2540.20 mm, GmbH & Co. KG Neumann-Neander-str. 6–8, Dueren, Germany) and visualized by a UV lamp with the wavelength fixed to 254 nm and/or with a solution of KMnO_4_ (1%). Molecular weights were determined by ESI-MS (Micromass ZMD 2000) and the values are reported as [M+H]^+^. IR spectra were registered with a Spectrum 100 FT-IR Spectrophotometer (PerkinElmer, Waltham, MA, USA) and the main band is reported as cm^−1^. Melting points were determined by a Stuart melting point apparatus. ^1^H-NMR and ^13^C-NMR were registered on the VXR-200 Varian spectrometer at 200 MHz and 400 MHz, using TMS (Tetramethylsilane) as an internal standard. The chemical shift of each signal is expressed as units δ (ppm) relative to the signal of deuterated solvents used (DMSO-*d_6_* and CDCl_3_). The following abbreviations are used to designate multiplicity and assignment: s (singlet), d (doublet), t (triplet), m (multiplet), dd (double doublet), BZT (benzothiazole), and Ar (Aryl). UV spectrophotometric analysis were conducted on a UV-Vis spectrophotometer (Shimadzu UV-2600, Columbia, MD, USA) or on a Life Science UV/VIS spectrophotometer (Beckman Coulter, DU^®^530, Single Cell Module, Beckman Coulter s.r.l., Via Roma, 108–Palazzo F1, Centro Cassina Plaza 20,060-Cassina De’Pecchi, Milano, Italy). Evaluations of antiproliferative activity were carried out with a Beckman Coulter^®^ Z2. Pro-apoptosis activity was assessed using a cytofluorimeter Muse^®^ Cell Analyzer (Merck-Millipore, Burlington, MA, USA) and kit Muse^®^ Annexin V & Dead Cell Reagent (Merck-Millipore Cat. No. MCH100105).

### 3.2. Chemistry

#### 3.2.1. Synthesis of Benzo[d]thiazole-2-carbohydrazide (2)

To a solution of benzo[d]thiazole-2-etylcarboxylate 1 (500 mg, 2.41 mmol) in EtOH (15 mL) in a 50 mL round bottomed flask, a 25% *w*/*w* solution of hydrazine hydrate (1.43 mL, 7.23 mmol) was added. The mixture was heated to reflux condition for 1 h. After completion, when the spot of the starting material was no longer visible on TLC, the solution was allowed to warm to room temperature. The needle-shaped crystal solid formed during cooling was separated on a Gooch filter and washed several times with cold deionized water, and then dried in oven at 50 °C overnight. The pale yellow crystals yield was 92%. Analytical data agree with those reported in literature [30].

#### 3.2.2. General Procedure for the Synthesis of Compounds **BZTidr1–6**

An equimolar mixture of compound **1** (1 mmol) with the adequate aromatic aldehyde (1 mmol) in EtOH (5 mL) was heated at reflux condition for 3–8 h. The reaction was monitored by TLC. After completion, the resulting suspension was allowed to cool to room temperature, and then filtered. The obtained solid was finally recrystallized from hot EtOH, MeOH or iPrOH in order to obtain the final products with high purity grade.

#### 3.2.3. Chemical Properties of Compounds **BZTidr1–6**

(E)-N′-benzylidenebenzo[d]thiazole-2-carbohydrazide (**BZTidr1**): white solid, yield 86.1%. m.p. 227–230 °C; IR (cm^−1^): 3211.14, 1656.65, 1525.31, 1486.36, 1119.13; ^1^H-NMR (400 MHz, CDCl_3_): δ(ppm) 10.56 (s, 1H, -NH), 8.40 (s, 1H, -N=CH-), 8.12 (d, 1H, J = 8 Hz, BZT), 8.02 (d, 1H, J = 7.6 Hz, BZT), 7.83 (m, 2H, Ar), 7.60 (t, 1H, BZT), 7.54 (t, 1H, BZT), 7.44 (m, 3H, Ar); ^13^C-NMR (400 MHz, CDCl_3_): δ(ppm) 162.8 (-CONH), 155.8 (BZT), 152.5 (BZT), 150.0 (-N=CH-), 137.06 (BZT), 133.2 (Ar), 131.0 (Ar), 128.8 (2C, Ar), 128.0 (2C, Ar), 127.1 (2C, BZT), 124.3 (BZT), 122.5 (BZT). ESI-MS [M+H]^+^: calculated for C_15_H_11_N_3_OS, 282.06; found, 282.12.

(E)-N′-(4-hydroxybenzylidene)benzo[d]thiazole-2-carbohydrazide (**BZTidr2**): pale yellow solid, yield: 76.7%. m.p. > 280 °C; IR (cm^−1^): 3322.75, 3297.42, 1648.42, 1603.60, 1576.36, 1512.78, 1129.01; ^1^H-NMR (400 MHz, DMSO-*d_6_*): δ(ppm) 12.44 (S, 1H, -NH), 9.99 (s, 1H, -OH), 8.55 (s, 1H, -N=CH-), 8.25 (d, 1H, J = 7.2 Hz, BZT), 8.17 (d, 1H, J = 7.6 Hz, BZT), 7.61 (m, 2H, BZT), 7.56 (d, 2H, J = 8.4 Hz, Ar), 6.84 (d, 2H, J = 8.8 Hz, Ar); ^13^C-NMR (400 MHz, DMSO-*d_6_*): δ(ppm) 164.5 (-CONH-), 160.3 (BZT), 156.3 (Ar), 153.2 (BZT), 151.3 (-N=CH-), 136.5 (BZT),129.7 (2C, Ar), 127.7 (BZT), 127.5 (Ar), 125.4 (BZT), 124.5 (BZT), 123.5 (BZT), 116.2 (2C, Ar). ESI-MS [M+H]^+^: calculated for C_15_ H_11_N_3_O_2_S, 298.06; found, 297.89.

(E)-N′-(4-methoxybenzylidene)benzo[d]thiazole-2-carbohydrazide (**BZTidr3**): pale yellow solid, yield: 78.6%. m.p. 208–215 °C; IR (cm^−1^): 3281.26, 2969.77, 1661.67, 1601.07, 1509.24, 1124.53; ^1^H-NMR (400 MHz, DMSO-*d_6_*): δ(ppm) 12.54 (s, 1H, -NH), 8.62 (s, 1H, -N=CH-), 8.27 (d, 1H, J = 7.6 Hz, BZT), 8.19 (d, 1H, J = 7.6 Hz, BZT), 7.69 (d, 2H, J = 8.8 Hz, Ar), 7.64 (m, 2H, BZT), 7.04 (d, 2H, J = 7.6 Hz, Ar), 3.82 (s, 3H, -OCH_3_); ^13^C-NMR (400 MHz, DMSO-*d_6_*): δ(ppm) 163.8 (-CONH-), 161.1 (BZT), 155.8 (Ar), 152.6 (BZT), 150.3 (-N=CH-), 135.9 (BZT), 128.9 (2C, Ar), 127.2 (BZT), 127.0 (BZT), 126.4 (Ar), 123.9 (BZT), 123.0 (BZT), 114.3 (Ar), 55.2 (-OCH_3_). ESI-MS [M+H]^+^: calculated for C_16_H_13_N_3_O_2_S, 312.07, found, 312.28.

(E)-N′-(3,4-dihydroxybenzylidene)benzo[d]thiazole-2-carbohydrazide (**BZTidr4**): yellow solid, yield: 80.3%. m.p. > 280 °C; IR (cm^−1^): 3522.93, 3230.15, 3160.64, 1653.33, 1509.01, 1243.18; ^1^H-NMR (400 MHz, DMSO-*d_6_*): δ(ppm) 12.41 (s, 1H, -NH), 9.45 (s, 1H, -OH), 9.31 (s, 1H, -OH), 8.65 (s, 1H, -N=CH-), 8.25 (d, 1H, J = 7.6Hz, BZT), 8.17 (d, 1H, J = 8 Hz, BZT), 7.62 (m, 2H, BZT), 7.25 (d, 1H, j = 2 Hz, Ar), 6.91 (d, 1H, J = 7.6 Hz, Ar), 6.78 (d, 1H, J = 8 HZ, Ar); ^13^C-NMR (400 MHz, DMSO-*d_6_*): δ(ppm) 164.5 (-CONH-), 156.2 (BZT), 153.2 (BZT), 151.5 (BZT), 148.9 (Ar), 146.2 (Ar), 136.5(BZT), 127.7 (BZT), 127.5 (BZT), 125.9 (Ar), 124.5 (BZT), 123.5 (BZT), 121.5 (Ar), 116.1 (Ar), 113.3 (Ar). ESI-MS [M+H]^+^: calculated for C_15_H_11_N_3_O_3_S, 314.05, found, 313.83.

(E)-N′-(3-hydroxy-4-methoxybenzylidene)benzo[d]thiazole-2-carbohydrazide (**BZTidr5**): yellow solid, yield: 72.2%. m.p. 252–256 °C; IR (cm^−1^): 3468.62, 3281.67, 3070.35, 2844.00, 1679.65, 1274.83, 1131.47; ^1^H-NMR (400 MHz, DMSO-*d_6_*): δ(ppm) 12.468 (s, 1H, -NH), 9.36 (s, 1H, -OH), 8.50 (s, 1H, -N=CH-), 8.24 (dd, 1H, J_1_ = 8 Hz, J_2_ = 0.8 Hz, BZT), 8.17 (dd, 1H, J_1_ = 7.2 Hz, J_2_ = 0.8 Hz, BZT), 7.62 (m, 2H, BZT), 7.27 (d, 1H, J = 2 Hz, Ar), 7.06 (dd, 1H, J_1_ = 8.4 Hz, J_2_ = 2 Hz, Ar), 6.98 (d, 1H, J = 8.4 Hz, Ar), 3.80 (s, 3H, -OCH_3_); ^13^C-NMR (400 MHz, DMSO-*d_6_*): δ(ppm) 163.9 (-CONH-), 155.8 (BZT), 152.7 (BZT), 150.6 (-N=CH-), 150.2 (Ar), 146.9 (Ar), 136.0 (BZT), 127.2 (BZT), 127.0 (BZT), 126.7 (Ar), 124.00 (BZT), 123.0 (BZT), 120.8 (Ar), 112.3 (Ar), 111.8 (Ar), 55.5 (-OCH_3_). ESI-MS [M+H]^+^: calculated for C_16_H_13_N_3_O_3_S, 328.07, found, 328.55.

(E)-N′-(3-methoxy-4-hydroxybenzylidene)benzo[d]thiazole-2-carbohydrazide (**BZTidr6**): bright yellow solid, yield: 84.0%. m.p. 232–237 °C; IR (cm^−1^): 3514.66, 3428.57, 3227.28, 2801.58, 1658.39, 1588.36, 1518.58, 1285.53,1268.26, 1128.38; ^1^H-NMR (400 MHz, DMSO-*d_6_*): δ(ppm) 12.47 (s, 1H, -NH), 9.63 (s, 1H, -OH), 8.54 (s, 1H, -N=CH-), 8.25 (dd, 1H, J_1_ = 7.6 Hz, J_2_ = 1.2 Hz, BZT), 8.18 (dd, 1H, J_1_ = 8 Hz, J_2_ = 1 Hz, BZT), 7.66 (m, 2H, BZT), 7.30 (d, 1H, J = 2 Hz, Ar), 7.09 (dd, 1H, J_1_ = 8.2 Hz, J_2_ = 1.8 Hz, Ar), 6.85 (d, 1H, J = 8 Hz, Ar), 3.83 (s, 3H, -OCH_3_); ^13^C-NMR (400 MHz, DMSO-*d_6_*): δ(ppm) 164.5 (-CONH-), 156.3 (BZT), 153.2 (BZT), 151.5 (-N=CH-), 149.9 (Ar), 148.5 (Ar), 136.4 (BZT), 127.7 (BZT), 127.5 (BZT), 125.8 (Ar), 124.49 (BZT), 123.5 (Ar), 123.1 (BZT), 116.0 (Ar), 109.6 (Ar), 56.0 (-OCH_3_). ESI-MS [M+H]^+^: calculated for C_16_H_13_N_3_O_3_S, 328.08, found, 328.09.

#### 3.2.4. General Procedure for the Synthesis of Compounds **BZTcin1–5**

Each cinnamic acid derivative (1.4 mmol) was suspended in 5 mL of CH_3_CN in a 50 mL round bottomed flask and cooled to 0 °C. WSC·HCl (1.4 mmol) was added portionwise followed by HOBt (1.0 mmol); the mixture was stirred for 30 min. The ice bath was removed and 2-aminobenzo[d]thiazole (1.5 mmol) was added to the solution. The reaction mixture was heated at 55–60 °C overnight. The solvent was removed under reduced pressure and the resulting solid was taken up in EtOAc (25 mL). The organic phase was in turn washed with HCl 0.5 M (3 × 10 mL), H_2_O (3 × 10 mL) and brine (3 × 10 mL). The aqueous phases were again extracted with EtOAc, if necessary. The collected organic phases were dried over anhydrous Na_2_SO_4_, filtered and concentrated. The solid recovered was then purified through column chromatography (EtOAc:Hexane or DCM:MeOH as eluents).

#### 3.2.5. Chemical Properties of Compounds **BZTcin1–5**

N-(benzo[d]thiazol-2-yl)cinnamamide (**BZTcin1**): white solid, yield: 47%. m.p. 229–234 °C; IR: 3127.87, 2950.53, 1682.85, 1598.17, 1547.83, 1268.27, 1160.53; ^1^H-NMR (400 MHz, DMSO-*d_6_*): δ(ppm) 12.58 (s, 1H, -NH), 7.99 (dd, 1H, J_1_ = 8 Hz, J_2_ = 0.8 Hz, BZT), 7.78 (d, 1H, J = 15.8 Hz, H_trans_), 7.75 (d, 1H, J = 8 Hz, BZT), 7.65 (m, 2H, Ar_ortho_), 7.45 (m, 4H, 2H BZT + 2H Ar_meta_), 7.30 (t, 1H, Ar_para_), 6,95 (d, 1H, J = 15.8 Hz, H_trans_); ^13^C-NMR (400 MHz, Acetone-*d_6_*): δ(ppm) 163.8 (BZT), 158.00 (-CONH-), 149.2 (BZT), 143.7 (-COC=C-), 134.6 (Ar), 132.4 (BZT), 130.4 (Ar), 129.0 (2C, Ar), 128.2 (2C, Ar), 126.0 (BZT), 123.6 (BZT), 121.4 (BZT), 120.80(-COC=C-), 119.2 (BZT). ESI-MS[M+H]^+^: calculated for C_16_H_12_N_2_OS, 281.07, found, 281.20.

(E)-N-(benzo[d]thiazol-2-yl)-3-(4-hydroxyphenyl)acrylamide (**BZTcin2**): yellow solid, yield: 35.2%. m.p. 275–27 8°C; IR: 3159.83, 2952.45, 1665.24, 1579.67, 1543.50, 1513.56, 1264.26, 1151.30; ^1^H-NMR (400 MHz, DMSO-*d_6_*): δ(ppm) 12.44 (s, 1H, -NH), 10.07 (s, 1H, -OH), 7.97 (d, 1H, J = 8 Hz, BZT), 7.73 (d, 1H, J = 8 Hz, BZT), 7.68 (d, 1H, J = 15.6 Hz, H_trans_), 7.49 (d, 2H, Ar), 7.42 (t, 1H, BZT), 7.29 (t, 1H, BZT), 6.84 (d, 2H, J = 8.4 Hz, Ar), 6.73 (d, 1H, J = 15.6 Hz, H_trans_); ^13^C-NMR (400 MHz, DMSO-*d_6_*): δ(ppm) 165.7 (BZT), 160.5 (-CONH), 159.8 (Ar), 149.4 (BZT), 143.1 (-COC=C-), 132.3 (BZT), 130.5 (2C, Ar), 126.3 (BZT), 125.7 (Ar), 123.5 (BZT), 122.0 (BZT), 120.6 (BZT), 117.0 (-COC=C-), 116. 5 (2C, Ar). ESI-MS[M+H]^+^: calculated for C_16_H_12_N_2_O_2_S, 297.06, found, 297.07.

(E)-N-(benzo[d]thiazol-2-yl)-3-(4-methoxyphenyl)acrylamide (**BZTcin3**): opaque white solid, yield: 50%. m.p. 260–262 °C; IR: 3167.69, 3061.20, 2955.63, 1678.16, 1596.83, 1544.58, 1514.39, 1257.36, 1157.94; ^1^H-NMR (400 MHz, DMSO-*d_6_*): δ(ppm) 12.49 (s, 1H, -NH), 7.97 (dd, 1H, J_1_ = 7.8 Hz, J_2_ = 1 Hz, BZT), 7.74 (d, 1H, J = 8 Hz, BZT), 7.73 (dd, 1H, J = 14.6 Hz, H_trans_), 7.60 (dd, 2H, J_1_ = 6.8 Hz, J_2_ = 2 Hz, Ar), 7.41 (t, 1H, BZT), 7.30 (t, 1H, BZT), 7.02 (dd, 2H, J_1_ = 6.8 Hz, J_2_ = 2 Hz, Ar), 6.82 (d, 1H, J = 15.6 Hz, H_trans_), 3.80 (s, 3H, -OCH_3_); ^13^C-NMR (400 MHz, DMSO-*d_6_*): δ(ppm) 164.8 (BZT), 161.7 (-CONH-), 158.61 (Ar), 149.2 (BZT), 143.4 (-COC=C-), 132.1 (BZT), 130.4 (2C, Ar), 127.2 (Ar), 126.5 (BZT), 123.9 (BZT), 122.1 (BZT), 120.9 (BZT), 117.2 (-COC=C-), 115.1 (2C, Ar). ESI-MS[M+H]^+^: calculated for C_17_H_14_N_2_O_2_S, 311.08, found, 311.15.

(E)-N-(benzo[d]thiazol-2-yl)-3-(3,4-dihydroxyphenyl)acrylamide (**BZTcin4**): yellow solid, yield: 27.1%. m.p. 261–268 °C (decomposition); IR: 3230.15, 3160.64, 1653.33, 1509.01, 1243.18, 1260.30, 1148.75, 1109.72; ^1^H-NMR (400 MHz, DMSO-*d_6_*): δ(ppm) 12.43 (s, 1H, -NH), 9.60 (s, 1H, -OH), 9.28 (s, 1H, -OH), 7.96 (d, 1H, J = 7.6 Hz, BZT), 7.73 (d, 1H, J = 7.6 Hz, BZT), 7.59 (d, 1H, J = 15.6 Hz, H_trans_), 7.42 (t, 1H, BZT), 7.28 (t, 1H, BZT), 7.05 (s, 1H, Ar), 6.95 (d, 1H, J = 8 Hz, Ar), 6.78 (d, 1H, J = 8 Hz, Ar), 6.65 (d, 1H, J = 15.6 Hz, H_trans_); ^13^C-NMR (400 MHz, DMSO-*d_6_*): δ(ppm) 164.9 (BZT), 158.7 (-CONH-), 149.1 (2C, Ar), 146.2 (BZT), 144.3 (-COC=C-), 132.1 (BZT), 126.5 (BZT), 126.1 (Ar), 123.9 (Ar), 122.3 (BZT), 122.1 (BZT), 120.9 (Ar), 116.3 (-COC=C-), 115.7 (Ar), 114.6 (Ar). ESI-MS[M+H]^+^: calculated for C_16_H_12_N_2_O_3_S, 313.06, found, 313.22.

(E)-N-(benzo[d]thiazol-2-yl)-3-(3-hydroxy-4-methoxyphenyl)acrylamide (**BZTcin5**): opaque white solid, yield: 29%. m.p. 268–273 °C; IR: 3310.91, 3062.02, 1686.51, 1611.10, 1541.48, 1510.70, 1258.78, 1151.85, 1127.29; ^1^H-NMR (400 MHz, DMSO-*d_6_*): δ(ppm) 12.47 (s, 1H, -NH), 9.33 (s, 1H, -OH), 7.97 (d, 1H, J = 7.6 Hz, BZT), 7.73 (d, 1H, J = 8 Hz, BZT) 7.63 (d, 1H, J = 15.6 Hz, H_trans_), 7.42 (t, 1H, BZT), 7.29 (t, 1H, BZT), 7.08 (m, 2H, Ar), 6.98 (d, 1H, J = 8.8 Hz, Ar), 6.71 (d, 1H, J = 15.6 Hz, H_trans_), 3.80 (s, 3H, -OCH_3_); ^13^C-NMR (400 MHz, DMSO-*d_6_*): δ(ppm) 164.8 (BZT), 158.6 (-CONH-), 150.7 (BZT), 149.2 (Ar), 147.3 (Ar), 143.9 (-COC=C-), 132.1 (BZT), 127.5 (BZT), 126.5 (Ar), 123.9 (BZT), 122.1 (BZT), 122.1 (BZT), 120.9 (-COC=C-), 116.8 (Ar), 114.0 (Ar), 112.6 (Ar), 56.1 (-OCH_3_). ESI-MS[M+H]^+^: calculated for C_17_H_14_N_2_O_3_S, 327.07, found, 326.91.

### 3.3. In Vitro Biological Assays

#### 3.3.1. DPPH Test

The procedure used was adapted from Wang et al. [31]. Briefly, 0.750 mL of a methanolic solution of each tested compound was treated with 1.5 mL of DPPH solution (0.004%) in MeOH. Samples were stored in the dark at room temperature for 30 min, and the absorbance value was read with a spectrophotometer at 517 nm, which is the maximum peak of absorbance relative to the DPPH radical absorption spectra. The values obtained were fitted into Equation (1) and converted to inhibition percentage (%) of the radical:*DPPH inhibition percentage (%)* = 100 × [1 − A1/A0](1)
where A1 and A0 were, respectively, the absorbance recorded with and without the sample, which are directly proportional to DPPH residual concentration in the sample and DPPH concentration in the control (0.750 of MeOH and 1.5 mL of DPPH solution). All compounds were first screened at a 1 mg/mL concentration; compounds which performed an inhibition of the radical >90% were further investigated in order to determine their IC_50_ value (expressed as µg/mL). A linear regression was performed to calculate the sample concentration able to eliminate 50% of DPPH free radicals.

#### 3.3.2. FRAP Assay

The FRAP test (Ferric Reducing Antioxidant Power) is based on a colorimetric reaction and evaluates the ability of a sample of reducing ferric ions to ferrous ion while complexed with TPTZ (2,4,6-trispyridil-s-triazine). The FRAP solution was prepared from 0.1 M acetate buffer (pH 3.6), 10 mmol/L of TPTZ in 40 mM HCl, and 20 mM solution of FeCl_3_, with a 10:1:1 ratio, as described [32]. Then, 1.9 mL of this solution was incubated with 0.1 mL of the sample solution at 37 °C for 10 min. The absorbance was read with a UV-Vis spectrophotometer at a 593 nm wavelength, relative to the maximum peak of absorbance of Fe^2+^-TPTZ complex. Trolox^®^ was used as a standard and the results are expressed as µmolTE/g.

#### 3.3.3. Photo-Protection Activity

##### Evaluation of Filtering Parameters of Formulation

The synthesized compounds were incorporated at a concentration of 3% in a typical oil-in-water sunscreen formulation, to evaluate the filtering parameters, and have the following ingredient composition:

INCI (International Nomenclature Cosmetic Ingredients): aqua, glycerin, Euxyl PE 9010, xanthan gum, Tribehenin PEG-20 Esters, cetearyl alcohol, dicaprylyl carbonate, PPG-26-Buteth-26 (and) PEG-40 Hydrogenated Castor Oil (and) water, C12-C15 Alkyl Benzoate, and Sodium hydroxide (sol. 10%).

As previously reported [16] the protocol indicates to spread 32.5 mg of each formulation on 25 cm^2^ PMMA plates with the fingers. Each formulation was tested in triplicate, and five measurements were recorded for each dish, following a 15–30 min incubation in the dark at room temperature. UV transmittance data was recorded using a UV-VIS spectrophotometer in the range 290–400 nm. The blank plate, SPF, UVA-PF and critical wavelength were obtained by SPF Calculator software (version 2.1), Shimadzu, Milan, Italy) [16].

##### Photostability Study

To determine the photostability, the previously reported method [16] was adopted, which consists in preparing each plate as mentioned above and subjecting it to solar irradiation quantified as the minimum dose of UVA equivalent to an effective radiant exposure of erythema. The protocol stipulates that, before and after exposure to sunlight, the transmission of the sun protection layer in the range of 290–400 nm is recorded.

Using Equations (2) and (3) below, the residual percentage of SPF in vitro (% SPFeff.) and UVA-PF (% UVA-PFeff.) have been obtained, which, if greater than or equal to 80, can classify a filter as photostable.
%SPF_eff_. = in vitro SPF_after_/in vitro SPF_before_ × 100(2)
%UVA − PF_eff_. = UVA − PF_after_/UVA − PF_before_ × 100(3)

#### 3.3.4. Cell Growth Inhibition Assays

Cell growth inhibition assays were carried out using human melanoma Colo-38 cells [33,34,35] and skin HaCat keratinocytes cells [15,21,33,34,35,36]. Human melanoma Colo-38 cells were kindly provided by Dr. Patrizio Giacomini (Laboratory of Immunology, Regina Elena Institute, Rome, Italy). HaCat cells were kindly provided by Istituto Zooprofilattico Sperimentale della Lombardia e dell’Emilia Romagna, Brescia, Italy. Cell lines were seeded using concentrations of 40,000 cells/mL and 25,000 cells/mL, respectively. Colo-38 cells were maintained in RPMI 1640, supplemented with 10% fetal bovine serum (FBS), penicillin (100 Units mL^−1^), and streptomycin (100 µg mL^−1^). HaCat cells were maintained in DMEM, supplemented with 10% fetal bovine serum (FBS), penicillin (100 Units mL^−1^), streptomycin (100 µg mL^−1^) and glutamine (2 mM); the incubation was performed at 37 °C in a 5% CO_2_ atmosphere. Tested compounds were dissolved in MeOH/DMSO (1:1) to obtain 50 mM stock solutions and diluted before cell treatment in MeOH 100%. The tested compounds were added at serial dilutions to the cell cultures (from 0.1 to 100 µM) and incubated for 24 h further. Cells were then harvested, suspended in physiological solution, and counted with a Z2 Coulter Counter (Coulter Electronics, Hialeah, FL, USA). The cell number/mL and the IC_50_ values were determined when untreated cells were in the log phase of cell growth. Cells treated with vehicles and untreated cells were placed in each plate.

#### 3.3.5. Pro-Apoptotic Activity (Annexin V Method)

Apoptosis assays were performed with a Muse^®^ Cell Analyzer instrument (Merck Millipore, Burlington, MA, USA) and the kit on Colo-38 and HaCat cell lines according to the manufacturer’s protocol. Cells were treated and exposed to the compounds for 24 h. This procedure is based on the detection of PS (PhosphatidylSerine) exposed on the external membrane of apoptotic cells by Annexin V. Four populations of cells can be distinguished using this assay: live, early apoptotic, late apoptotic and dead cells. The MUSE cell analyzer can discriminate “early” from “late” apoptosis since “early apoptotic cells” are Annexin V-positive and 7-AAD-negative, while “late apoptotic cells” are positive to both Annexin V and 7-AAD (7-aminoactinomycin D). The DNA intercalating molecule 7AAD (7-aminoactinomycin D) is used as an indicator of cell membrane integrity, since it binds DNA only in cells undergoing late apoptosis/death stage, when the membrane integrity is lost [21,37,38,39]. Muse Annexin V and Dead Cell reagent were used to dilute the cells (1:1). After 20 min of incubation between cell samples and the kit at room temperature in the dark, the samples were analyzed using 0.01% Triton X-100 as positive control [40]. The data were acquired and recorded using the Annexin V and Dead Cell software module (Millipore, Billerica, MA, USA).

## 4. Conclusions

Following the recent trend of scientific research in finding new molecular entities endowed with two or more synergic biological activities and continuing our efforts on the investigation of privileged structures, the aim of this work was to obtain and evaluate two series of benzothiazole derivatives as potential multifunctional agents for the treatment of skin-related diseases. For this reason, the crucial aspect of this work was to assess the ability of those compounds to act as antioxidants, UV-filters and selective antiproliferative agents. Of the 11 compounds obtained, six (**BZTidr1–6**) incorporated the acyl-hydrazone linker which was previously applied to other heterocycles with encouraging results. The remaining five (**BZTcin1–5**) derive from the design of a new linker structure, mimicking the acyl-hydrazone and retaining π conjugation. Each compound was first tested in vitro for the evaluation of an antioxidant profile through DPPH and FRAP tests. For both series, compounds carrying a cathecolic moiety (**BZTidr4** and **BZTcin4**) emerged as the best candidates, with a great DPPH radical scavenging ability, which was also confirmed by the low micromolar IC_50_ values and antioxidant activity in the FRAP test (slightly superior to reference standard Trolox^®^). The results confirm that 3,4-dihydroxyphenyl is one of the best moieties for this specific biological activity. In addition, **BZTidr6**, bearing a 3-methoxy-4-hydroxyphenyl moiety, shows an interesting antioxidant capacity. As far as the linker role is concerned, it can be hypothesized that the acyl-hydrazone moiety may favor the antioxidant capacity of the benzothiazole derivatives, as values obtained from both DPPH and FRAP tests are slightly higher for the **BZTidr** series, as compared to the respective **BZTcin** derivative bearing the same substituent pattern. Moreover, the evaluation of filtering parameters of the benzothiazole derivatives, incorporated in a suitable o/w emulsion, yielded interesting results; in particular, **BZTcin2** and **BZTcin4** (which shares a *p*-hydroxyl moiety on the aromatic ring) show an excellent broad-spectrum filtering capacity towards both UVB and mostly UVA radiation, which highlights the potential capacity of these multifunctional agents as candidates for organic sunscreen UV filters, as they are able to shield from a wide range of wavelength of the UV spectra. Introduction of -OH groups in the p-position seems to shift the UV absorption capacity from UVB to UVA, as proven by the slightly higher UVA/UVB ratio for those two candidates. In addition, both compounds were found to be photostable as the values of SPFeff.% and UVAPF-eff.% were above the threshold value of 80. To support the desired multifunctional profile, all the derivatives were tested in vitro for their antiproliferative activity on Colo-38, a selected human melanoma cell line, as all **BZTidr** compounds were found to be inactive or weakly active. However, we observed some interesting data for **BZTcin2**, **BZTcin4** and **BZTcin5**, which inhibited Colo-38 cell proliferation by 50% at low-micromolar concentrations. Furthermore, on the HaCat cell line, all compounds tested did not show any significant activity up to 100 µM concentration, with the exception of **BZTcin4** and **BZTcin5**, which were still highly selective toward the tumoral cell line. These results underline and confirm the biological potential of the benzothiazole scaffold. The effect of the application of two distinct types of linkers led to the enhancement of different biological activities; in fact, the best results in terms of antioxidant capacity were reached by the **BZTidr** series, while the **BZTcin** series were superior in terms of UV-filtering and antiproliferative activity. Taking everything into account, we identify **BZTcin4** as the best candidate for a multifunctional profile. This molecule is indeed endowed with a good UV-filtering capacity, coupled with a high antioxidant activity, and was also able to exert an in vitro antiproliferative activity towards a selected human melanoma cell line with an interesting IC_50_ value. Although further studies are required in order to assess an in vivo efficacy of the aforementioned formulations, the outcomes of this study are interesting and support further investigations on multifunctional candidates based on the benzothiazole scaffold.

## Data Availability

Data is contained within the article and Appendix A.

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
