# Peer review of "Design, Synthesis and Evaluation of New Multifunctional Benzothiazoles as Photoprotective, Antioxidant and Antiproliferative Agents"

_molecules, 2022, doi:10.3390/molecules28010287_

Round 1

Reviewer 1 Report

The author has developed multifunctional benzothiazole derivatives as photoprotective, antioxidant, and antiproliferative agents. The introduction provides a good explanation and rationale behind developing these derivatives. The author has synthesized 11 new derivatives and evaluated them for antioxidant activity, filtering parameters, photoprotective activity, photostability, and antiproliferative activity against melanoma. In my recommendation, the manuscript can be accepted after minor revision.

Comments:

1)    Provide a brief explanation about DPPH and FRAP tests and why it was performed to determine antioxidant activity.

2)    What is the solar formulation?

3)    In Table 3, please delete HCT116 if it was not utilized for evaluation

4)    In SI, please show proton NMR integration.

Author Response

REVIEWER 1

Comments and Suggestions for Authors

The author has developed multifunctional benzothiazole derivatives as photoprotective, antioxidant, and antiproliferative agents. The introduction provides a good explanation and rationale behind developing these derivatives. The author has synthesized 11 new derivatives and evaluated them for antioxidant activity, filtering parameters, photoprotective activity, photostability, and antiproliferative activity against melanoma. In my recommendation, the manuscript can be accepted after minor revision.

Comments:

1)    Provide a brief explanation about DPPH and FRAP tests and why it was performed to determine antioxidant activity.

As requested by the reviewer, an introductory section on antioxidant testing with explanation has been included in section 2.2.

In recent years, antioxidants and their therapeutic potential in the prevention and modulation of oxidative damage have assumed a central role, and this has prompted increasing interest in researching new molecules, natural and synthetic, to counteract the course of disorders associated with excess free radicals in the body.

There are several in vitro investigation methods for measuring antioxidant capacity: the different methods allow the analysis of the reactivity of molecules toward different types of radicals, depending on the nature and mechanism of action of the antioxidant species.

To carry out the evaluation of the antioxidant properties of benzothiazole derivatives, the compounds of the two series were assayed by the following two methods: the DPPH test and the FRAP test (Table 1).

The DPPH assay is one of the simplest and fastest in vitro methods for evaluating the antioxidant power of pure extracts or compounds, based on the depletion of the 2,2-diphenyl-1-picrylhydrazyl radical (DPPHË™) by an antioxidant molecule with scavenger activity. This is an extremely sensitive test that can detect the activity of active ingredients present in very low concentrations.

The FRAP (Ferric Reducing Antioxidant Power) test allows the measurement of the ability of antioxidants to reduce ferric ion to ferrous. This method represents one of the simplest, quickest and least expensive methods for the determination of antioxidant properties in vitro.

2)    What is the solar formulation?

A solar formulation (or sun care formulation, or sunscreen) is to be understood as a formulation that contains ingredients with a fine balance between SPF efficacy, formulation stability, and product aesthetics.

In the text solar has been replaced with sun care or sunscreen.

3)    In Table 3, please delete HCT116 if it was not utilized for evaluation

As suggested by the reviewer, HCT116 was deleted from Table 3.

4)    In SI, please show proton NMR integration.

As requested by the reviewer, all integrations of the NMR spectra have been added.

In addition, the manuscript has been thoroughly carefully revised.

Reviewer 2 Report

molecules-2101179 numbers The article titled ''Design, synthesis and evaluation of new multifunctional benzothiazoles as photoprotective, antioxidant and antiproliferative agents'' is generally well designed and flowing in harmony. But there are some major shortcomings. Relevant revision requests are listed below.

-d6 should be written in subscript and italics.

-Cytotoxicity, apoptosis and DPPH studies should be added as positive control.

-Mass spectra are missing in the supplementary material. It should be added.

- The digits after the comma in the activity tables should be uniform.

Author Response

REVIEWER 2

Comments and Suggestions for Authors molecules-2101179 numbers

The article titled ''Design, synthesis and evaluation of new multifunctional benzothiazoles as photoprotective, antioxidant and antiproliferative agents'' is generally well designed and flowing in harmony. But there are some major shortcomings. Relevant revision requests are listed below.

-d6 should be written in subscript and italics.

As suggested by the reviewer, d6 was written in subscript and italics.

-Cytotoxicity, apoptosis and DPPH studies should be added as positive control

As requested by the reviewer, the IC50 value for the DPPH test of caffeic acid used as a positive control was added in Table 1.

Regarding the positive control information in cytotoxicity and apoptosis tests, information has been included in the text in Section 2.4 (lines 329-331) and in Section 2.5 in the caption of Figure 2, respectively.

-Mass spectra are missing in the supplementary material. It should be added.

As suggested, mass spectra were added.

- The digits after the comma in the activity tables should be uniform.

As rightly pointed out by the auditor, all the digits after the comma in the activity tables have been unified.

Round 2

Reviewer 2 Report

The article can be accepted in this form.

Author Response

Thank you so much.